# Did You Mean...? Confidence-based Trade-offs in Semantic Parsing

**Elias Stengel-Eskin**[*]
UNC Chapel Hill
esteng@cs.unc.edu

**Benjamin Van Durme**
Johns Hopkins University
vandurme@jhu.edu

## Abstract

We illustrate how a calibrated model can help balance common trade-offs in task-oriented parsing. In a simulated annotator-in-the-loop experiment, we show that well-calibrated confidence scores allow us to balance cost with annotator load, improving accuracy with a small number of interactions. We then examine how confidence scores can help optimize the trade-off between usability and safety. We show that confidence-based thresholding can substantially reduce the number of incorrect low-confidence programs executed; however, this comes at a cost to usability. We propose the DidYouMean system (cf. Fig. 1) which better balances usability and safety by rephrasing low-confidence inputs.

## 1 Introduction

Task-oriented dialogue systems (Gupta et al., 2018; Cheng et al., 2020; Semantic Machines et al., 2020) represent one path towards achieving the long-standing goal of using natural language as an API for controlling real-world systems by transforming user requests into executable programs, i.e. translating natural language to code. Central to the systems' success is the ability to take rational actions under uncertainty (Russell and Norvig, 2010). When model confidence is low and the system is unlikely to succeed, we would prefer it defer actions and request clarification, while at high confidence, clarification requests may annoy a user. Relying on model confidence requires it to be well-correlated with accuracy, i.e. it requires a *calibrated* model.

Recent work has focused on the calibration of semantic parsing models. Specifically, Stengel-Eskin and Van Durme (2022) benchmarked the calibration characteristics of a variety of semantic parsing models, finding some of them to be well-calibrated, especially on parsing for task-oriented dialogue. Given the relatively well-calibrated nature of these

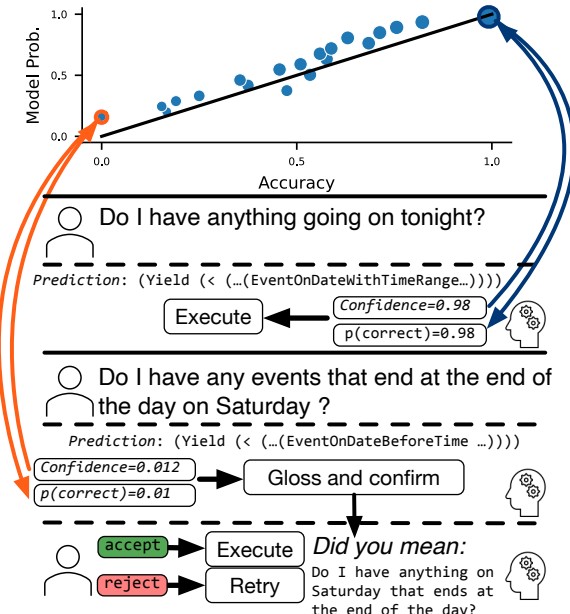

Figure 1: The DidYouMean system. At high confidences, we simply execute the predicted parse. At low confidences, DidYouMean rephrases the query based on the predicted program and asks a user to confirm the paraphrase. The program is executed if the user accepts.

models, we first examine how they could be used in an annotation interface, with a view to balancing the trade-off between *annotation cost* and *correctness*. We simulate a human-in-the-loop (HITL) experiment where high-confidence tokens are automatically annotated and low-confidence tokens trigger a dialogue with an oracle annotator who either picks the correct token from a top-K list or manually inserts it. With a small number of interactions we substantially boost annotator accuracy.

A similar trade-off exists between *usability* and *safety* in task-oriented user interfaces. We examine how sequence-level model confidence scores can be used to balance this trade-off by reducing the number of incorrect programs executed while also minimizing the number of follow-up user interactions and their cognitive burden. We find that thresholding outputs based on model confidence

---

[*]Work done while at Johns Hopkins University.

(i.e. rejecting outputs falling below a tuned threshold) reduces the number of incorrect programs executed by 76% compared to the baseline. However, this comes at a cost to usability, as roughly half the correctly-predicted parses are also rejected. To strike a balance between safety and usability, we introduce the DidYouMean system (cf. Fig. 1), which rephrases the input conditioned on the predicted parse and asks users to confirm the accuracy of the paraphrase. In a user study, we obtain an 36% improvement in usability over the thresholded system while maintaining a 58% reduction in the number of incorrect programs executed.

## 2 Related Work

Our experiments in Section 4 involve a predictive model for human-in-the-loop coding: similar models have been integrated into IDEs, e.g. Chen et al. (2021). DidYouMean relates to the interactive semantic parsing domain (Li and Jagadish, 2014; Chaurasia and Mooney, 2017; Su et al., 2018), where humans are included in the semantic parsing loop. In this domain, Yao et al. (2019) introduce a confidence-based interactive system in which a parsing agent can ask users for clarification. Our work follows in this spirit, but asks the user to confirm a parse rather than generating questions for the user to answer; we also operate in a different parsing domain, preventing us from using Yao et al.'s system as a baseline. DidYouMean also relates broadly to selective prediction, where a model is expected to abstain from making decisions at low confidence (Chow, 1957; El-Yaniv et al., 2010; Varshney et al., 2022; Xin et al., 2021; Whitehead et al., 2022). Our system extends selective prediction's setting by including a human-in-the-loop. Finally, DidYouMean shares a motivation with Fang et al. (2022), who introduce a method for reliably summarizing programs. Their work provides post-hoc action explanations while we focus on resolving misunderstandings *before* execution.

## 3 Methods

**Datasets** Our data is drawn from the SMCalFlow (Semantic Machines et al., 2020) task-oriented dialogue dataset, which contains Lisp-like programs (cf. Appendix A). We follow the same preprocessing as Platanios et al. (2021), and use the SMCalFlow data splits given by Roy et al. (2022): 108,753 training, 12,271 validation, and 13,496 testing dialogue turns.

**Models** We use MISO (Zhang et al., 2019a,b), a well-calibrated model from Stengel-Eskin and Van Durme (2022). Rather than predict the SMCalFlow surface form, including syntactic tokens like parentheses, MISO directly predicts the underlying execution graph. The graph can deterministically be "de-compiled" into its surface form, and vice-versa. The fact that MISO predicts an underlying graph makes it attractive for applications which require confidence scores, as it only predicts content tokens (i.e. functions, arguments) rather than structure markers, like parentheses. For details on MISO's architecture, see Zhang et al. (2019b) and Stengel-Eskin et al. (2022).

For token confidence estimation, we use the maximum probability across the output vocabulary at each timestep. This has been shown to be a relatively robust confidence estimator in classification (Hendrycks and Gimpel, 2016; Varshney et al., 2022). For sequence-level scores, we follow Stengel-Eskin and Van Durme (2022) and take the minimum over token-level confidence scores.

## 4 Human-in-the-Loop Simulation

Production datasets like SMCalFlow are constantly evolving as new functionalities are added. The expensive and time-consuming nature of annotating data can be mitigated by the use of predictive parsing models which suggest speculative parses for new utterances. However, the model's output can be incorrect, especially given out-of-distribution inputs. We need to ensure that annotators are not introducing errors by overly trusting the model.

If the model is well-calibrated, we can use its confidence to reduce such errors. For example, we can alert annotators to low confidence predictions and ask them to intervene (Lewis and Gale, 1994). Using a threshold, we can prioritize time or correctness: a higher threshold would result in more annotator-model interactions, decreasing the speed but increasing program correctness (reducing the need for debugging) while a lower threshold would increase speed but also lower the accuracy.

Since we do not have access to expert SMCalFlow annotators, we simulate an oracle human-in-the-loop (HITL) annotator who always provides a correct answer by using the gold annotations provided in the dataset. Specifically, for a given input, we decode the output tokens of a predicted program $o_0, \ldots o_n$ normally as long as predictions are confident (above a given threshold). If at time $t$ the confidence $p(o_t)$ falls the threshold, we attempt to

match the decoded prefix $o_0, \ldots, o_{t-1}$ to the gold prefix $g_0, \ldots g_{t-1}$. If the prefixes do not match, we count the example as incorrect. If they do match, we replace $o_t$ with $g_t$, the gold prediction from our oracle annotator, and continue decoding. We consider three metrics in this experiment: (1) The exact match accuracy of the decoded programs (higher is better). (2) The percentage of total tokens for which we have to query an annotator (lower is better, as each query increases annotator load). (3) The percentage of uncertain tokens (below the threshold) for which the gold token $g_t$ is in the top 5 predictions at timestep $t$. Here, higher is better, as selecting a token from a candidate list is typically faster than producing the token.

**Results and Analysis** Fig. 2 shows our three metrics as the threshold is increased in increments of 0.1. We see that accuracy grows exponentially with a higher threshold, and that the percentage of tokens for which an annotator intervention is required grows at roughly the same rate. The exponential growth reflects the distribution of token confidences, with most tokens having high confidence. Finally, we see that while at low confidence, most tokens must be manually inserted, the rate at which they are chosen from the top 5 list rapidly increases with the threshold. Thus, the increased number of annotator interactions required at higher thresholds may be offset by the fact that many of these interactions are a choice from the top-5 list.

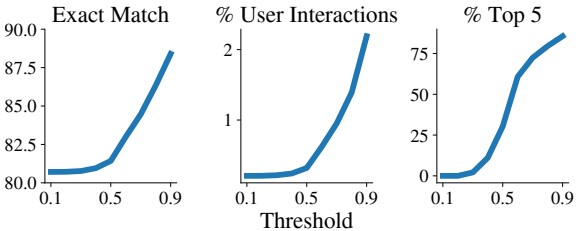

Figure 2: Simulated annotator-in-the-loop results across increasing confidence thresholds.

## 5 User Correction via DidYouMean

Section 4 showed that token-level confidence scores can be used to balance speed and correctness in an annotation interface. We see a similar trade-off between safety and usability in user interfaces using semantic parsing models. Here, we define safety as rejecting unsuccessful programs *before executing them*. This strict definition is motivated by physical domains: imagine that rather than controlling a digital assistant, a user is guiding a robot via language commands (e.g. Winograd (1972); Lynch and Sermanet (2020); Stengel-Eskin et al.

(2021); Lynch et al. (2022); Nair et al. (2022)). In this setting, actions may have irreversible consequences, so determining safety before execution is key. Safety considerations need to be balanced with usability of the system: an unplugged agent would be very safe but unusable. To increase usability, an agent might make follow-up requests to a user, like asking for clarification or confirmation. The types of requests the agent makes have varying cognitive load on the user: for example, providing confirmation takes less effort than rephrasing.

We measure how well we can reject incorrect programs *before* executing them. Following past work in selective prediction (El-Yaniv et al., 2010; Varshney et al., 2022, i.a.), we measure success by coverage and risk, as well as F score w.r.t. program correctness. Coverage is the percentage of inputs for which a program is executed and risk is the percentage of executed programs that were *incorrect*. Precision is inverse risk, and recall is the percentage of correct programs which were accepted. In addition to F1, we consider F0.5, which upweights precision (safety) by a factor of 2. A low-coverage, low-risk system may be safer but have more false negatives, i.e. reject more correct programs, decreasing its usability. A high-coverage, high-risk system is more usable at the cost of false positives, i.e. executing incorrect programs. We do not commit to setting an optimal threshold for this trade-off, since it is task-specific.

We consider 3 systems. As a baseline, we consider a system that executes everything it predicts (**accept**); this will result in the highest-possible coverage, but also high risk. We can also use MISO's calibrated nature to improve safety outcomes by tuning a sequence-level confidence threshold for rejecting programs (**tuned**). We tune on the full validation set using F1; we explore the range $[0.0, 1.0)$ in increments of $0.01$, finding $0.40$ to be optimal. Finally, we introduce the **DidYouMean** system for filtering low-confidence programs. For a given utterance, DidYouMean shows the user a paraphrase of the input; the user then decides to accept the parse based on this paraphrase. This allows correctly-predicted low-confidence programs to be accepted and executed, while reducing the user load: making a binary choice to accept a paraphrase is a receptive task, while rephrasing an instruction is a more costly productive task. Details of DidYouMean's components are given below.

**Glossing Model** Since users are typically unfamiliar with formats like Lisp, we need to present the user with a natural language paraphrase – or *gloss* – of the candidate parse. To train a glossing model, we modify Roy et al. (2022)'s seq2seq BenchCLAMP semantic parsing framework: rather than using the user utterance with the previous turn's context $(\mathcal{U}_0, \mathcal{A}_0, \mathcal{U}_1)$ as input and a program $\mathcal{P}$ as output, we take the context and *program* $(\mathcal{U}_0, \mathcal{A}_0, \mathcal{P})$ as the input and user instruction $\mathcal{U}_1$ as the output. We use BART-large (Lewis et al., 2020) and fine-tune it on the SMCalFlow training split. See Appendix A for model details.

**DidYouMean System** When a low-confidence parse is detected, DidYouMean triggers a dialogue with the user in order to recover some usability over simply rejecting all low-confidence parses. Fig. 1 shows the system workflow. DidYouMean shows the original utterance $\mathcal{U}_1$ and the gloss $\hat{\mathcal{U}}^*$ to the user, who determines whether they are identical or not. If they accept the gloss, we optionally re-parse the gloss $\hat{\mathcal{U}}^*$ rather than the original utterance $\mathcal{U}_1$; this can remove typos and other idiosyncracies. We call this the **re-parsed** setting, while choosing the original prediction $\hat{\mathcal{U}}$ is the **chosen** setting. We predict that allowing users to accept and reject glosses will improve the balance between safety and usability (i.e. F1) over the threshold system by allowing them to accept correct low-confidence parses. In other words, adding human interaction will allow us to achieve a balance which cannot be attained given the tradeoffs resulting from thresholding.

**User Study** We conduct a static user study of DidYouMean with examples from the SMCalFlow validation set. We sample 100 MISO predictions with a confidence below 0.6 (to ensure that the set contains a number of mistakes). This sample is stratified across 10 equally-spaced bins with 10 samples per bin. MTurk annotators were shown the dialogue history, the user utterance, and the gloss, and asked to confirm that the gloss matched the utterance's intent. The template and instructions can be seen in Appendix B. We obtained three judgments from three different annotators per example.

**Annotation Statistics** In total eight annotators participated in the task, with four completing the majority of tasks. For each example, all three annotators agreed on $79\%$ examples, indicating the task is well-formulated. For the remaining $21\%$, we use the majority decision to accept or reject.

| Setting | Cov. ↑ | Risk ↓ | FP ↓ | F1 ↑ | F0.5 ↑ |
|---|---|---|---|---|---|
| Accept | 1.00 | 0.67 | 67 | 0.50 | 0.38 |
| Tuned | 0.32 | 0.50 | 16 | 0.49 | 0.50 |
| Chosen | 0.68 | 0.54 | 37 | 0.61 | 0.51 |
| Re-parsed | 0.68 | 0.41 | 28 | 0.66 | 0.62 |

Table 1: Coverage, risk, number of false positives (FP), and F measures for accepting correct parses and rejecting incorrect parses.

After majority voting, annotators accepted $68/100$ glosses and rejected 32.

**Results** Table 1 shows the results of the user study. In addition to standard selective prediction metrics like coverage (the percentage of inputs for which a program is executed) and risk (the percentage of executed programs that are incorrect) we report the number of false positives (incorrect programs executed) and F1 and F0.5 scores. Tuning a threshold yields better safety outcomes than accepting everything, with lower risk. However, this safety comes at a cost to the usability of the system; a coverage of only 0.32 indicates that only $32\%$ of inputs have their programs executed. The "tuned" system's low usability is reflected in the F1 and F0.5 scores, which balance precision and recall. The "chosen" system, while better in F1, is comparable to the "tuned" system in F0.5, which takes both usability and safety into account but prioritizes safety at a 2:1 ratio. Users are able to recover some usability (as measured by coverage) in this setting but also add to the risk, which is higher for "chosen" than "tuned". The number of incorrect programs executed increases when glosses are chosen (as compared to the tuned threshold). When the accepted glosses are re-parsed, we see a shift back towards a system favoring safety, with fewer incorrect programs being executed than in the "chosen" setting; this is reflected in a lower risk score. For both F1 and F0.5, the "re-parsed" system best balances usability and safety.

These results show that a calibrated model can be used with a threshold to greatly improve safety, reducing the number of incorrect programs accepted by $76\%$. DidYouMean allows users to recover some low-confidence programs by accepting and rejecting programs based on their glosses, resulting in the best aggregated scores. Note also that the threshold was tuned on F1 score on the entire dev set. This means that the F1 performance of that tuned system is as high as possible for confidence-threshold-based system. Thus, DidYouMean achieves a balance outside what can be

achieved by tuning: simply increasing the threshold would decrease safety and result in a lower F1 than the current threshold of 0.40.

## 6 Selection Study

In Section 5 we presented the DidYouMean system, which allowed users to confirm or reject a potential gloss. The gloss was chosen from a larger set of candidates; this naturally raises the question of whether users can directly choose a gloss from the candidate list, rather than accepting or rejecting a single gloss at a time. Here, we examine to what extent it is helpful to users to *choose* glosses from a list of options. We take a sample of validation programs for SMCalFlow, stratified across 10 evenly-spaced confidence bins from 0.0 to 1.0. Note that this differs from the setting in Section 5, where the maximum bin was 0.6. We then present the top predictions decoded with nucleus sampling (Holtzman et al., 2019); the number of predictions depends on the confidence of the model. Specifically, we set a cutoff of 0.85 and add predictions until the sum of the sequence-level confidence scores is greater than the cutoff; this could be achieved by a single very confident prediction or multiple low-confidence predictions. To reduce the load on the annotators, we limit the number of predictions seen to a max of 10, even if the cutoff is not reached.

Annotators were asked to choose one of the top predictions or to manually rewrite the query if none of the predictions were adequate; the chosen or rewritten query was then re-parsed and compared to the gold parse. Further annotation details are given in Appendix C. Of the 100 examples sampled, annotators manually rewrote 7 and chose from the top-$k$ list for the other 93. Ignoring the rewritten examples, 39 model predictions were incorrect and 54 were correct; by choosing glosses, annotators correct 5 incorrect predictions. However, they also inadvertently changed 4 correct predictions to incorrect. Figure Fig. 3 shows the exact match accuracy before and after annotators selected glosses at each confidence bin. At low confidence, we see very minor increases on the order of a single program being corrected. At high confidence, annotators generally have only one or two options, and are able to choose the correct one, resulting in similar performance to nucleus decoding. However, at medium confidence, annotators often choose the wrong gloss, leading

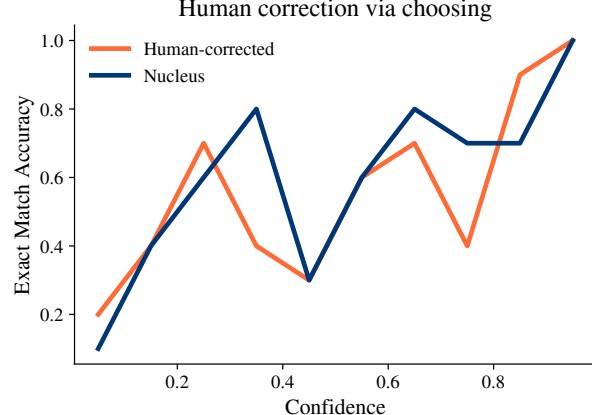

Figure 3: Selection experiment. Annotators sometimes select accurate glosses, but often have trouble deciding between seemingly invariant glosses, lowering performance especially at medium confidences.

to lower performance. Qualitatively, many incorrect choices are driven by glosses that appear to be paraphrases, but in fact correspond to subtly different Lisp programs. These results suggest that accepting/rejecting glosses is more promising than choosing them.

## 7 Conclusion

We examine two common trade-offs in semantic parsing, and how a well-calibrated model can be used to balance them. In Section 4 we illustrated how token-level model confidences could be used in a simulated HITL task-oriented parsing annotation task. Our experiments in Section 5 extended these results to sequence-level confidences and non-expert users; we found that model confidence could be used to improve the usability-safety trade-off and introduced DidYouMean, which improved usability by asking users to accept predictions.

## Limitations

Our study is limited by the models, datasets, and languages we consider. Firstly, we examine only English datasets, limiting the impact of our results. We also only consider one task-oriented parsing dataset, and focus on one model architecture.

We make several limiting assumptions in Section 4 and Section 5. Foremost amongst these is the assumption of access to an oracle annotator in Section 4; clearly, no such annotator exists. Our results may vary when real annotators are brought into the loop. For one, we do not know exactly how choosing from the top-k list will compare to insertion w.r.t. speed. We also do not know how

automation bias (Cummings, 2004) would affect the top-k list: given that the correct answer is often in the list, real annotators might overly rely on the list and prefer it to token insertion, resulting in incorrect programs.

The experiments in Section 5 rely on a glossing model to translate predicted programs into natural language (NL). We approached with a neural Lisp-to-NL model; this has several limitations. Neural text generation models often hallucinate outputs, i.e. generated glosses may not be faithful to their corresponding programs. Unlike Fang et al. (2022), who use a grammar-based approach for response generation, we do not assume access to a grammar but note that our method is compatible with grammar-based constraints. Our annotators in Section 5 face the additional challenge of interpreting and choosing glosses. SMCalFlow programs are nuanced and slight input variations can result in different programs. These nuances are often obscured by the glossing model, resulting in two different programs glossing to semantically equivalent utterances; we explore this further in Appendix C. Annotators might mistakenly accept glosses from incorrect programs or reject correct glosses; this would be difficult to address even with a faithful translation method.

## Acknowledgements

We would like to thank Anthony Platanios, Subhro Roy, Zhengping Jiang, Kate Sanders, Yu Su, and Daniel Khashabi for their feedback on an earlier draft. This work was supported by NSF #1749025, and an NSF Graduate Research Fellowship to the first author.

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

## A  Model and Data

**Data**  We use version 2.0 of SMCalFlow, which contains language inputs paired with Lisp-like programs. Fig. 4 shows an example input and output for SMCalFlow. Unlike many other task-oriented formalisms, SMCalFlow contains salience operators like `refer`, which allow for deixis in the input.

| Input | Output |
|---|---|
| Remove the even for the 23rd from the calendar. | `(Yield (UpdateCommitEventWrapper (UpdatePreflightEventWrapper (Event.id (singleton (QueryEventResponse.results (FindEventWrapperWithDefaults (EventOnDate (nextDayOfMonth (Today) 23L) (Event.subject_? (?~= "annual general meeting"))))))) (Event.start_? (DateTime.date_? (?= (MD 7L (May))))))))` |
| *I didn't find any events on Tuesday the 23rd.* | |
| No I meant the annual general meeting, change the dates from the 23rd to 7th may | |

Figure 4: Input-output example for SMCalFlow

**Input Representation**  We follow Stengel-Eskin et al. (2022) and Roy et al. (2022) in using the previous dialogue turn as input if available. Thus, each datapoint consists of an input $X = (\mathcal{U}_0, \mathcal{A}_0, \mathcal{U}_1)$ and an output program $\mathcal{P}$, where $\mathcal{U}_0$ is the previous user utterance (if it exists), $\mathcal{A}_0$ is an automatically-generated agent response to the previous utterance, and $\mathcal{U}_1$ is the current user utterance.

**Paraphrasing**  To find a good paraphrase of each predicted parse, we generate $N$ glosses $\hat{\mathcal{U}}_1, \ldots, \hat{\mathcal{U}}_N$ via beam search and take the one that yields the highest probability of the predicted parse $\hat{\mathcal{P}}$, i.e. $\hat{\mathcal{U}}^* = \text{argmax}_i \, P_{MISO}(\hat{\mathcal{P}}|\mathcal{U}_0, \mathcal{A}_0, \hat{\mathcal{U}}_i)$.

**Evaluating the glossing model**  Instead of evaluating the glossing model using string metrics such as BLEU (Papineni et al., 2002) which can be noisy, we choose to evaluate the output programs using cycle-consistency. Specifically, we evaluated trained models by glossing programs in the gold test set and then parsing those glosses with a fixed MISO parser. We explore T5-base, T5-large, BART-base, and BART-large architectures. All accuracy scores are reported in Fig. 5 along with the baseline accuracy obtained by parsing the gold test inputs. The best-performing gloss model is BART-large. Note that all glossing models outperform MISO without glosses. This can be explained by the fact that we gloss the inputs from the gold program, which we then evaluate on,

allowing for information leakage. We also hypothesize that the gloss model, having been trained on the entire dataset of utterances, averages out many annotator-specific idiosyncracies which may make inputs hard to parse. This result does *not* imply that glosses generated from *predicted* programs would yield better performance when parsed than the user input.

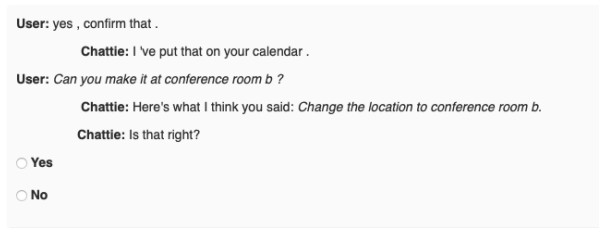

Figure 5: Exact match for programs parsed from glossed inputs from the SMCalFlow test set. Performance without glosses is overlaid in black.

## B   User study details

Fig. 6 shows the template for the confirmation HIT. The instructions asked users to read the paraphrase produced by the agent and determine whether the agent had correctly understood the user. Annotators

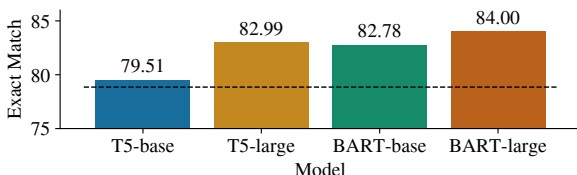

Figure 6: Template for the confirmation HIT.

were recruited from a list of qualified annotators on Amazon Mechanical Turk and paid $0.05 per example, averaging to roughly $16 per hour.

## C   Selection study details

The template for the selection study is shown in Fig. 7; each example gives the gloss as well as a rounded confidence score for the predicted program. Annotation was run with the same group of annotators as the experiments in Section 5; annotators were paid $0.11 per example, or about $16 per hour of annotation. Each example was annotated by a single annotator, all of whom had been vetted in a pilot annotation task. Annotators were instructed to help a robot named SvenBot, who had recently learned English and was not confident about its understanding, disambiguate between several options. The interface contained a text box where annotators could optionally manually re-write the input;

this was only to be done in cases where *none* of the options reflected the intended meaning of the original utterance.

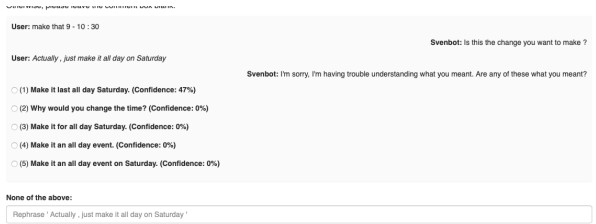

Figure 7: Template for the selection HIT.