# OpenReview forum: "Did You Mean...? Confidence-based Trade-offs in Semantic Parsing"
_EMNLP/2023/Conference — EMNLP 2023 Main_

### Official Review · Reviewer_k4cA · 2023-07-30

**Soundness:** 4

**Excitement:**

3: Ambivalent: It has merits (e.g., it reports state-of-the-art results, the idea is nice), but there are key weaknesses (e.g., it describes incremental work), and it can significantly benefit from another round of revision. However, I won't object to accepting it if my co-reviewers champion it.

**Paper Topic And Main Contributions:**

This work develops a system to use calibration of semantic parsers to reduce the amount of annotation needed to train these models. Specifically, the authors craft a simulated annotator-in-the-loop experiment which uses calibration scores to determine whether a specific query needs to be rephrased and checked with the user. This can then be used to reduce errors in system output, which the authors argue reduces some of the need for a high amount of annotated data. Specifically, the authors demonstrate that token-level confidence scores (by proxy of three metrics) can be used to balance usability and safety; the "DidYouMean" system allows low-confidence parses to have a second chance at resulting in a correct response by triggering a dialogue which rephrases the question back to the user. The work also presents a user study of the DidYouMean system which shows relatively high inter-annotator agreement, and also shows that ensuring safety comes at a high cost of usability, so a threshold will need to be used to effectively implement the system.

**Questions For The Authors:**

Question A: Why did you choose the three metrics you used in Section 4?

Question B: Who were the annotators in Section 5 and what specific content were they shown (were they also seen the original utterance and the gloss per how the DidYouMean system would actually be used)?

**Reasons To Accept:**

The work here is well-motivated and provides sufficient details and results for a short paper. The writing is clear and the experimentation is thorough. Efforts towards mitigating the need for expert annotators are valuable and this is an interesting application of calibration, being that the confidence score is used to filter high-risk predictions (which is possibly innovative for semantic parsing).

**Reasons To Reject:**

The human-in-the-loop experiments are simulated using an annotator who is always correct, as the authors did not hire expert annotators. The assumption that any given annotator is 100% accurate on even a simple task is flawed and this study would be more rigorous if (1) either real annotators were used, or (2) disagreement analysis was considered in the simulation, rather than relying on one simulated accurate annotator. It is also not clear why the three employed metrics were chosen, i.e. if their use is motivated by prior work or what information they capture.

**Reproducibility:**

2: Would be hard pressed to reproduce the results. The contribution depends on data that are simply not available outside the author's institution or consortium; not enough details are provided.

**Reviewer Confidence:**

2: Willing to defend my evaluation, but it is fairly likely that I missed some details, didn't understand some central points, or can't be sure about the novelty of the work.

**Typos Grammar Style And Presentation Improvements:**

The "Semantic Machines et al.," 2020 citation is not correct

Figure 1 is slightly hard to understand at first-- I think it would help if the graph went underneath the dialogue so that the reader's first inclination isn't to see where the orange arrow is going.

---

### Official Review · Reviewer_q8g5 · 2023-08-02

**Soundness:** 4

**Excitement:**

4: Strong: This paper deepens the understanding of some phenomenon or lowers the barriers to an existing research direction.

**Paper Topic And Main Contributions:**

This paper describes experiments in balancing safety vs. usefulness of a dialogue system that generates and executed code based on the input.  First, the authors determine confidence scores on the generated code and conduct simulated human-in-the-loop studies setting different confidence score thresholds.  The higher the confidence score, the fewer generated programs will execute.  Then, the authors developed the DidYouMean system which, for generated code below the confidence score threshold, will try to rephrase the user's query in an effort to better understand the ask and generate safer (higher confidence) code.  DidYouMean is evaluated using eight MTurk annotators.

**Reasons To Accept:**

This paper is very well-written and well-organized.  The experiments are sound and the results are reasonable.  Multiple annotators were employed with good inter-annotator agreement.  The Limitations section is thorough and addresses my concern about the use of machine-generated "glosses" in lieu of Lisp code presented to the annotators.

I think this paper will get other researchers thinking about generated response confidence and safety vs. usefulness.  I wonder if, in larger production systems, a user-specific confidence threshold couldn't be learned.

**Reasons To Reject:**

Some reviewers may not think this work is novel enough.

**Reproducibility:**

4: Could mostly reproduce the results, but there may be some variation because of sample variance or minor variations in their interpretation of the protocol or method.

**Reviewer Confidence:**

4: Quite sure. I tried to check the important points carefully. It's unlikely, though conceivable, that I missed something that should affect my ratings.

**Typos Grammar Style And Presentation Improvements:**

On line 046, the comma after "annotator" is unnecessary.

On line 289, you first mention four annotators but then say "All 3".  Should this be four, or is additional information needed to discuss a subset of three annotators?

In that same paragraph, it fits better with the formal style of conference papers to write numbers ten and under using words.

On line 301, I think "results" isn't necessary before "yields".

---

### Official Review · Reviewer_DWn7 · 2023-08-10

**Soundness:** 3

**Excitement:**

3: Ambivalent: It has merits (e.g., it reports state-of-the-art results, the idea is nice), but there are key weaknesses (e.g., it describes incremental work), and it can significantly benefit from another round of revision. However, I won't object to accepting it if my co-reviewers champion it.

**Paper Topic And Main Contributions:**

Topic:

The paper investigates the effects of model calibration in task-oriented parsing through a simulated annotator-in-the-loop experiment.

Main contributions:

- Shows how well-calibrated confidence scores can allow for balancing cost with annotator load, increasing accuracy through a small number of interactions.
- Confidence-based thresholding reduces the number of incorrect executions at a cost to usability, whereby applying the proposed system (DidYouMean) improves that balance by rephrasing low-confidence inputs.

**Questions For The Authors:**

How come real annotators were not brought into the loop, as indicated in the limitations section?

**Reasons To Accept:**

For the most part a well-written and well-rounded short-paper contribution on an intriguing topics which makes for a rather compelling read. The results are believable and well-presented.

**Reasons To Reject:**

The paper is at times a bit dense and would definitely benefit from an extra page to unpack some of the experiment setup and results discussion paragraphs. The content as it is may be a bit hard to follow for readers with whom the topic falls outside core areas of expertise.

**Reproducibility:**

3: Could reproduce the results with some difficulty. The settings of parameters are underspecified or subjectively determined; the training/evaluation data are not widely available.

**Reviewer Confidence:**

3: Pretty sure, but there's a chance I missed something. Although I have a good feel for this area in general, I did not carefully check the paper's details, e.g., the math, experimental design, or novelty.

---

### Official Review · Reviewer_11rs · 2023-08-11

**Soundness:** 3

**Excitement:**

3: Ambivalent: It has merits (e.g., it reports state-of-the-art results, the idea is nice), but there are key weaknesses (e.g., it describes incremental work), and it can significantly benefit from another round of revision. However, I won't object to accepting it if my co-reviewers champion it.

**Paper Topic And Main Contributions:**

This paper presents a new system for semantic parsing that rephrases low-confidence inputs and asks users to confirm the paraphrases. By leveraging the relatively cheap human signal (answering Yes/No questions), the system strikes a good usability-safety trade-off for semantic parsing.

**Questions For The Authors:**

A. Could you justify your current baseline selection better? Why are other HITL or automated baselines missing?

**Reasons To Accept:**

+ A new HITL system that improves task performance at a small cost of human involvement.

**Reasons To Reject:**

- Missing baselines: The paper presents a system that relies on human feedback (confirming rephrasing) and shows that system outperforms a naive system that accepts all predictions and a system with confidence thresholds tuned for F1. However, the paper does not compare to previous work (e.g., "Yao et al. (2019) introduce a confidence-based interactive system in which a parsing agent can ask users for clarification." in line 79) or other obvious baselines (e.g.,  a system with another model to select the most likely rephrase).

**Reproducibility:**

3: Could reproduce the results with some difficulty. The settings of parameters are underspecified or subjectively determined; the training/evaluation data are not widely available.

**Reviewer Confidence:**

1: Not my area, or paper was hard for me to understand. My evaluation is just an educated guess.

**Typos Grammar Style And Presentation Improvements:**

- Wrong format: Limitations should not be a numbered section

---

### Meta-Review · Area_Chair_uSLz · 2023-09-17

**Recommendation:** 4

**Metareview:**

This paper investigates the use of calibrated models to address two trade-offs for dialogue systems that generate and execute code based on user input: (1) cost/speed vs. accuracy in annotation, and (2) usability vs. safety in task-oriented user interfaces.

- The first question is addressed by showing the benefits of manually annotating only low confidence samples, although the experiments rely on a simulated annotator, which is always correct, thus limiting the generalizability of the results.

- For the second question, safety is viewed as rejecting an unsuccessful program before executing it, while usability is not clearly defined but is operationalized as the percentage of inputs for which a program is executed. Here, the paper shows that confidence-based thresholding reduces the number of incorrect executions at the cost of coverage, and that rephrasing a user's query for low confidence outputs yields safer (ie higher confidence) outputs. This system is evaluated in a user study with 8 MTurk annotators.

Overall, the paper represents an interesting focused contribution that is appropriate for a short paper.

I will note that it was not very clear to me what is gained by looking at these two problems jointly in a single short paper. I would recommend either addressing that point more directly, or focusing the paper on the second study.

---

### Decision · Program_Chairs · 2023-10-07

**Decision:**

Accept-Main

**Comment:**

This paper investigates the use of calibrated models to address two trade-offs for dialogue systems that generate and execute code based on user input: (1) cost/speed vs. accuracy in annotation, and (2) usability vs. safety in task-oriented user interfaces.

- The first question is addressed by showing the benefits of manually annotating only low confidence samples, although the experiments rely on a simulated annotator, which is always correct, thus limiting the generalizability of the results.

- For the second question, safety is viewed as rejecting an unsuccessful program before executing it, while usability is not clearly defined but is operationalized as the percentage of inputs for which a program is executed. Here, the paper shows that confidence-based thresholding reduces the number of incorrect executions at the cost of coverage, and that rephrasing a user's query for low confidence outputs yields safer (ie higher confidence) outputs. This system is evaluated in a user study with 8 MTurk annotators.

Overall, the paper represents an interesting focused contribution that is appropriate for a short paper.

I will note that it was not very clear to me what is gained by looking at these two problems jointly in a single short paper. I would recommend either addressing that point more directly, or focusing the paper on the second study.